# In Vitro and In Vivo Effects of the Combination of Polypurine Reverse Hoogsteen Hairpins against HER-2 and Trastuzumab in Breast Cancer Cells

**DOI:** 10.3390/ijms24087073

**Published:** 2023-04-11

**Authors:** Ester López-Aguilar, Patricia Fernández-Nogueira, Gemma Fuster, Neus Carbó, Carlos J. Ciudad, Véronique Noé

**Affiliations:** 1Department of Biochemistry and Physiology, School of Pharmacy and Food Sciences, Universitat de Barcelona (UB), 08028 Barcelona, Spain; 2Unit of Biophysics and Bioengineering, Department of Biomedicine, School of Medicine and Health Sciences, Universitat de Barcelona (UB), 08028 Barcelona, Spain; 3Department of Biochemistry and Molecular Biomedicine, School of Biology, Universitat de Barcelona (UB), 08028 Barcelona, Spain; 4Tissue Repair and Regeneration Laboratory (TR2Lab), Institut de Recerca i Innovació en Ciències de la Vida i de la Salut a la Catalunya Central (IrisCC), 08500 Vic, Spain; 5Department of Biosciences, Faculty of Sciences, Technology and Engineering, Universitat de Vic-Universitat Central de Catalunya (UVic-UCC), 08500 Vic, Spain; 6Institute of Biomedicine, Universitat de Barcelona (IBUB), 08028 Barcelona, Spain; 7Institute of Nanoscience and Nanotechnology, Universitat de Barcelona (IN2UB), 08028 Barcelona, Spain

**Keywords:** PPRH, HER-2, breast cancer, gene silencing, trastuzumab

## Abstract

Therapeutic oligonucleotides are powerful tools for the inhibition of potential targets involved in cancer. We describe the effect of two Polypurine Reverse Hoogsteen (PPRH) hairpins directed against the *ERBB2* gene, which is overexpressed in positive HER-2 breast tumors. The inhibition of their target was analyzed by cell viability and at the mRNA and protein levels. The combination of these specific PPRHs with trastuzumab was also explored in breast cancer cell lines, both in vitro and in vivo. PPRHs designed against two intronic sequences of the *ERBB2* gene decreased the viability of SKBR-3 and MDA-MB-453 breast cancer cells. The decrease in cell viability was associated with a reduction in *ERBB2* mRNA and protein levels. In combination with trastuzumab, PPRHs showed a synergic effect in vitro and reduced tumor growth in vivo. These results represent the preclinical proof of concept of PPRHs as a therapeutic tool for breast cancer.

## 1. Introduction

Modulation of gene expression is a strong technique used to reduce the synthesis of proteins associated with pathological processes. Numerous technologies for gene regulation based on nucleic acids are currently available, such as antisense oligodeoxynucleotides (aODNs), small interfering RNAs (siRNAs), or Triplex-Forming Oligonucleotides (TFOs).

In this work, we used a silencing tool developed in our laboratory, the Polypurine Reverse Hoogsten hairpins (PPRHs) [1]. These are DNA molecules formed by two polypurine antiparallel strands without modifications and linked by a thymidine loop of four or five Ts. The two DNA stretches form a hairpin structure by binding with each other through intramolecular reverse Hoogsteen bonds. These hairpins then bind to their polypyrimidine target sequences in a sequence-specific manner, located either in the coding [2] or template [3] DNA strands, via Watson-Crick bonds. Coding-PPRHs recognize their target sequence in the coding strand of the DNA, and since it has the same orientation and sequence as the mRNA, these hairpins can also bind to mRNA.

PPRHs present many advantages over other silencing tools. They are inexpensive compared to siRNAs and show higher stability because of their DNA structure. They are also effective at low concentrations, in the nanomolar range, and it has been demonstrated that they are not immunogenic [4]. 

PPRHs can decrease gene expression through different mechanisms [2,5], and they have been used in previous studies to inhibit different targets involved in cancer, such as DHFR, survivin, or telomerase [3]. Furthermore, PPRHs have also been demonstrated to be useful in other processes such as immunotherapy (acting on the SIRPalpha/CD47 and PD1-PDL-1 pathways) [6,7] and replication stress (Wee1 and CHK1) [8]. 

We decided to study the in vivo and in vitro effects of PPRHs against the *ERBB2*/*NEU* oncogene, which encodes for the human epidermal growth factor receptor 2 (HER-2). This tyrosine-kinase receptor participates in the regulation of cell proliferation, survival, and differentiation [9]. Overexpression of *ERBB2* occurs in several forms of cancer, such as breast, stomach, colon, ovary, lung, head and neck, uterine serous endometrial carcinoma, esophagus, bladder, and uterine cervix [9], and it also serves as a predictive biomarker. In this work, we focused on HER-2-positive breast cancer since the amplification of HER-2 occurs approximately in 15–20% of the cases and is correlated with a higher grade, a more aggressive phenotype, and a worse prognosis [10]. Treatment of HER-2-positive breast cancers includes the use of monoclonal antibodies such as trastuzumab or pertuzumab, kinase inhibitors, or different antibody-drug conjugates such as trastuzumab-deruxtecan or trastuzumab-emtansine [10,11]. Since the combination of trastuzumab or pertuzumab and a single-agent chemotherapeutic is the standard first-line treatment for most patients with HER-2-positive breast cancer, we chose to study the effects of PPRHs directed against *ERBB2* sequences in combination with trastuzumab, a more targeted HER-2-therapy than chemotherapy.

Thus, the aim of this work was to evaluate the efficacy of PPRHs against the *ERBB2* gene in HER-2-positive breast cancer cell lines, alone or in combination with the antibody trastuzumab, both in vitro and in vivo. We showed that two PPRHs designed against the *ERBB2* gene were able to silence the *ERBB2* gene and protein expression and cause a significant decrease in the viability of breast cancer cells in vitro and tumor growth in vivo, either alone or in combination with trastuzumab.

## 2. Results

We designed two PPRHs against the ERBB2 gene, Hp-Her2-I4-T (HpI4) and Hp-Her2-I6-C (HpI6), one of them directed against the template strand (HpI4) and the second against the coding strand (HpI6). Both PPRHs were directed against polypyrimidine stretches in two intronic regions of the ERBB2 gene (T). The two target sequences have either three or two purine interruptions. The complementary pyrimidine to those interruptions was included in the PPRHs at those sites to maintain perfect binding to the target sequence by Watson-Crick bonds without mismatches [12]. 

### 2.1. Effects of PPRHs on SKBR-3 Cells

We first evaluated the efficiency of transfection of the cationic liposome DOTAP in SKBR-3 cells. Cells were incubated for 24 h with a negative control PPRH labeled with the fluorescein (FAM)-liposome complex, and the internalization of the PPRH was analyzed by fluorescent microscopy and flow cytometry. The negative control PPRH was used instead of HpI4 and HpI6, directed against specific sequences of the ERBB2 gene, to avoid any decrease in cell viability caused by the potential silencing effect of these PPRHs on ERBB2 expression. As shown in Figure 1a, most cells were fluorescent 24 h post-transfection, confirming the ability of DOTAP to internalize the PPRH. In Figure 1b, the amount of fluorescence inside the cells quantified by flow cytometry is shown. At a concentration of 100 nM of PPRH transfected with 10 µM DOTAP, 90.1% of the cells were fluorescein-positive, with an X-mean of 32.2. 

Then, the effects on SKBR-3 cell viability were analyzed for both PPRHs designed against ERBB2 in a range of concentrations from 10 to 100 nM. As shown in Figure 1c, both PPRHs were effective at reducing cell viability in a dose-dependent manner. At 100 nM, HpI6 caused a decrease of almost 45% and, noteworthy, HpI4 dramatically reduced up to 90% the viability of SKBR-3 breast cancer cells at the same concentration. Next, the expression of HER-2 upon transfection of HpI4 and HpI6 was analyzed both at the mRNA and protein levels. As shown in Figure 1d, ERBB2 mRNA levels were decreased by 49% when transfecting HpI4 or by 60% for HpI6, respectively. These decreases in mRNA levels led to a reduction in the protein levels of the PPRH target gene ERBB2 (Figure 1e) of 77 +/− 4.5% for HpI4 and 67 +/− 3.8% for HpI6, respectively.

### 2.2. Effects of the Combination of PPRHs and Trastuzumab on SKBR-3 Cell Viability 

Next, we wanted to test the effect of the combination of both HpI4 and HpI6 with trastuzumab (TZ), a humanized monoclonal antibody against HER-2 used together with chemotherapy as a standard treatment in HER-2-positive breast cancer [13]. We first evaluated the cytotoxicity of TZ by itself in SKBR-3 cells. Cells were incubated with increased concentrations of the antibody, and cell viability was determined by the MTT assay. As shown in Figure 2a, we only observed approximately a 10–12% decrease in cell viability independent of the amount of antibody used in our conditions, which is consistent with its mechanism of action, which highly relies on the induction of antibody-dependent cell-mediated cytotoxicity (ADCC) [14]. 

To evaluate a possible additive or synergistic effect of both PPRHs in combination with TZ, SKBR-3 cells were transfected with either HpI4 or HpI6 in the presence of 100 µg/mL of TZ, a concentration that only causes a 10–12% reduction in cell viability (Figure 2a). The reduction in cell viability was analyzed over a range of PPRH concentrations of 30–100 nM. As shown in Figure 2b,c, in all cases, the combination of each PPRH with the antibody was more effective than either the PPRH or TZ alone. When 30 nM of HpI4 or HpI6 were combined with 100 µg/mL of TZ, a 62% and 35% decrease in viability were observed, respectively. The cytotoxic effect increased to 74% (HpI4) and 51% (HpI6) when using 50 nM of PPRH, and to 90% (HpI4) and 60% (HpI6) at a concentration of 100 nM. 

To analyze the effect of these combinations, the CompuSyn software (v 1.0) was used to calculate the Combination Index (CI). With this approximation, the combinations of HpI4 and TZ showed a synergistic effect, especially at the concentration of 100 nM, with a CI value of 0.51 (Table 1). For HpI6, the combination index also indicated a synergistic effect, with CI values of 0.66 and 0.60 at 30 and 50 nM PPRH, respectively, whereas for 100 nM, the effect was nearly additive (CI = 0.93) (Table 1), even though the effects at 50 and 100 nM of this PPRH on cell viability were not statistically significant (Figure 2c).

### 2.3. In Vitro Effects of PPRHs in MDA-MB-453 Cells 

To explore the in vivo effects of the PPRHs designed against the ERBB2 gene in chick embryo chorioallantoic membrane (CAM) assays, we selected the MDA-MB-453 cell line, which overexpresses HER-2 and has already been used for this kind of assay [15]. We first evaluated the efficiency of DNA transfection in MDA-MB-453 cells using DOTAP and the negative control PPRH labeled with FAM by fluorescent microscopy (Figure 3a). Additionally, the cellular uptake of the PPRH in these cells was determined by flow cytometry by measuring the percentage of fluorescent cells and their mean fluorescence intensity 24 h after transfection of the fluorescent PPRH (Figure 3b). As shown in Figure 3a, DOTAP at 30–40 µM was able to internalize the fluorescent PPRH at a concentration of 300–400 nM in MDA-MB-453 cells, and more than 95% of the cells were fluorescent 24 h after transfection (Figure 3b). 

Once we had set up the conditions for the transfection of MDA-MB-453 cells, we evaluated the cytotoxic effect of HpI4 and HpI6 in this cell line (Figure 3c). Since at least 10^6^ cells were needed for the in vivo assays, we increased the number of cells and concentrations of DOTAP and PPRHs accordingly. Cell viability of MDA-MB-453 cells (330,000 cells/well) was evaluated at a range of PPRH concentrations from 250 to 400 nM after 3 days of incubation. At a concentration of 250 nM, the effect on cell viability was about a 33% decrease for HpI4 and 18% for HpI6. Cytotoxicity augmented to 56% and 46%, respectively, when a concentration of 400 nM PPRH was used.

Furthermore, we analyzed ERBB2 mRNA expression after transfection of MDA-MB-453 cells with the two above-mentioned PPRHs. HpI4 caused a 52% decrease in ERBB2 mRNA levels, a reduction that reached about 75% when HpI6 was used in the transfection (Figure 3d). 

### 2.4. Effects of the Combination of PPRHs and Trastuzumab on MDA-MB-453 Cell Viability

Following the same approach as in SKBR-3 cells, we studied the effect of the combination of both PPRHs with trastuzumab on MDA-MB-453 cell viability. As previously observed for the SKBR-3 cell line, the effect of 100 µg/mL of TZ on MDA-MB-453 cells was a reduction of about 11% in cell survival (Figure 4a). Then, three different concentrations (250 nM, 300 nM, and 400 nM) of both PPRHs were evaluated in combination with 100 µg/mL of the antibody. As previously observed, the effect on cell viability followed a dose-response effect for both HpI4 and HpI6, reaching a decrease of 56% and 44%, respectively, when cells were transfected with 400 nM of PPRHs (Figure 4b,c). Then, combinations of PPRHs at these same concentrations with 100 µg/mL of TZ were analyzed. In all cases, the effects of the combinations on cell viability were greater than the effects of the PPRHs alone. For HpI4, using a concentration of 400 nM and 100µg/mL of TZ, a 64% decrease in cell viability was observed (Figure 4b). The calculated Combination Index (CI) showed moderate synergism for 250 and 300 nM of the hairpin and a slight effect for the combination with 400 nM of HpI4 (Table 2). In the case of HpI6, the effect on viability in cells treated with 100 µg/mL of trastuzumab and 250, 300, or 400 nM of PPRH was about 26%, 42%, and 56% reduction, respectively (Figure 4c). In that case, CI indicated a slight synergic effect for 250 nM PPRH and a moderate effect of synergism for the combinations with 300 and 400 nM of HpI6 (Table 2).

The determination of HER-2 receptor protein levels showed a decrease of 35 +/− 2.1% and 28 +/− 1.3% in MDA-MB-453 cells after 48 h of incubation with 300 nM of HpI4 or HpI6, respectively. That effect increased to 65 +/− 3.8% and 52 +/− 3.0% when the combination of both PPRHs with 100 µg/mL of TZ was used (Figure 4d). 

### 2.5. In Vivo Assays in the CAM Model 

In order to evaluate the effects of PPRHs against in vivo HER-2 tumor growth, CAM assays were performed. Both PPRHs had similar effects on cell viability in MDA-MB-453 cells; HpI6 was chosen for the first in vivo determinations. 330,000 cells/well were transfected with 400 nM of HpI6 in combination with 40 µM DOTAP. Forty-eight hours later, cells were harvested, resuspended in PBS supplemented with CaCl_2_ and MgCl_2_, and inoculated in the chicken chorioallantoic membrane according to the steps depicted in Figure 5a. Seven days after inoculation, eggs were opened, and tumors extracted. As shown in Figure 5b,c, a concentration of 400 nM of HpI6 caused a dramatic 88% decrease in tumor size compared to the controls transfected with a negative control. PPRH tumors derived from either controls or HpI6-transfected cells were analyzed by Western blot to determine HER-2 protein levels. As shown in Figure 5d, the levels of HER-2 protein were decreased by 57% in tumor samples derived from cells transfected with Hp16. 

To assay the in vivo effect of the combinations of both PPRHs with trastuzumab, we chose a concentration of PPRHs of 300 nM and 100 µg/mL of TZ. As shown in Figure 6 and in Table 3, in these conditions, we observed a decrease in tumor size of 44% in samples derived from cells transfected with HpI4 and of 52% in samples derived from HpI6 transfection. Trastuzumab caused a reduction in tumor weight of 33%, whereas its combination with HpI4 and HpI6 led to a decrease of 70% and 78%, respectively. Furthermore, the levels of HER-2 protein were analyzed by Western blot in pools of tumors corresponding to each experimental group included in Figure 6a. As it can be observed in Figure 6b, in tumors derived from cells treated with TZ, the levels of HER-2 protein were 10% of those in the control, whereas in tumors derived from cells transfected with HpI4 and HpI6, the decrease in HER-2 protein levels was 95% and 70%, respectively. Finally, in tumors derived from cells treated with the combinations of PPRH and TZ, the levels of the HER-2 protein showed a decrease of almost 100% for HpI4 and 80% for HpI6, respectively.

## 3. Discussion

In this work, we describe the capability of the PPRH technology to produce gene silencing of the *ERBB2* gene expression in breast cancer cells, causing a significant decrease in in vitro cell viability and in vivo tumor growth. 

We have previously shown that PPRHs are a valuable tool for gene silencing of relevant cancer targets such as dihydrofolate reductase, survivin, BCL2, TOP1, mTOR, MDM2, and MYC [3,16]. Here, we designed two PPRHs against two different intronic polypyrimidine sequences present in the *ERBB2* gene. The existence of a few purine interruptions (2–3) in the DNA target was solved in the design by introducing their corresponding complementary nucleotide in the PPRH, as previously shown in Rodríguez L. et al. [12]. This design ensures the maximum specificity of the binding of the PPRHs to the sequence of the target. Transfection with the cationic liposome DOTAP, widely used in eukaryotic cells, allowed the internalization of fluorescent PPRH in different breast cancer human cell lines and was used as the transfection agent for all the experiments performed here.

We demonstrated that the cytotoxic effect observed in SKBR-3 and MDA-MB-453 cells that overexpress HER-2 was due to the inhibition of *ERBB2* gene expression, both at the mRNA and protein levels. In SKBR-3 cells, the inhibition of *ERBB2* expression was already achieved at low concentrations (10–30 nM) of the PPRHs, when normally the working concentration of PPRHs for gene silencing is 100 nM. Other approaches based on the usage of different types of oligonucleotides had already been applied to silence the expression of the *ERBB2* gene. In this direction, antisense oligonucleotides were used in combination with chemotherapy as an approach for the treatment of breast cancer [17] or as a strategy to sensitize resistance cells to chemotherapy [18]. The antiproliferative effects of *ERBB2*-specific antisense oligonucleotides have been described in HER-2 overexpressing breast cancer cells in a range of concentrations of 1–20 µM [19]. On the other hand, the small interfering RNA (siRNA) technology has also been tested to reduce HER-2 protein levels in cancer cells. In 2004, Choudhury et al. [20] described for the first time the use of siRNA to silence the *ERBB2* gene in different cell lines, leading to growth inhibition and apoptosis. Furthermore, siRNAs against the *ERBB2* gene have also shown promising results in overcoming resistance to trastuzumab and lapanitib in HER-2-positive cancer cell lines [21]. Triplex-Forming Oligonucleotides (TFOs) directed against promoter regions of the *ERBB2* gene can reduce *ERBB2* mRNA levels by 42% and protein levels by 59% in MCF-7 cells [22], and a TFO targeting the coding region of *ERBB2* has been shown to increase apoptosis and reduce tumor growth without a significant effect on HER-2 expression. The authors concluded that the effect observed for this specific TFO was dependent on the induction of DNA damage and independent of HER-2 cellular function [23].

Taken all together, therapeutic oligonucleotides are a potential alternative to antibodies and small molecule inhibitors to silence the expression of the *ERBB2* gene and could overcome some of the inconvenients of the standard treatments for HER-2 positive breast cancers. It is worth noting that PPRHs are more stable, less immunogenic, and less expensive than siRNAs [4]. PPRHs are effective and stable without the need for chemical modifications in their synthesis, as opposed to antisense oligonucleotides. When compared to TFOs, PPRHs bind to their target sequences at lower concentrations and cause a higher effect on cell viability, thus indicating a higher affinity and greater effectiveness [12].

Since trastuzumab is representative of the standard treatment of HER-2 positive breast tumors, we studied its effect in combination with the PPRHs against the *ERBB2* gene. In this sense, the rationale for a combination of trastuzumab with either chemotherapy such as capecitabine or paclitaxel, or with tyrosine kinase inhibitors such as lapatinib, or more recently with a dual inhibitor of both mTOR and PI3K, has proved to be a more potent approach for HER-2-positive breast cancer treatment than trastuzumab alone [24]. Nowadays, the combined therapy of trastuzumab, pertuzumab, and taxane is the standard for treating metastatic breast cancer [25]. Our results indicated that PPRHs could be a potential tool in combination therapies with trastuzumab since their gene silencing mechanism contributed to increasing the effect on cell viability with respect to the antibody alone. The limited effect of trastuzumab on cell viability in the in vitro assays could be explained by its need for the induction of antibody-dependent cell-mediated cytotoxicity (*ADCC*) [14]. Nevertheless, the combination of trastuzumab and both PPRHs against *ERBB2* revealed a certain degree of synergism depending on the concentration of PPRH tested, confirming the potential applicability of such combinations as therapeutic strategies for HER-2-positive breast cancer.

Finally, the combination of PPRHs with trastuzumab could represent a reasonable option to reduce the cost of cancer care, which is currently rising and might not be sustainable soon. Specifically, the antibody cost represents $54,000 for a year’s worth of treatment, and the estimated trastuzumab-related costs as a single agent in an adjuvant regime can reach $70,000 per patient [26].

The in vivo experiments using the CAM model represent the pre-clinical proof of concept for using the specific PPRHs against *ERBB2* to inhibit tumor growth. We had previously shown the effectiveness of a PPRH against *BIRC5* (survivin) in a xenograft model using prostate cancer cells. In this model, tumor growth was reduced by 30–50% after the administration of the PPRH for three weeks, and the reduction in tumor growth correlated with a decrease in the protein target levels of 50–70% [5]. In the present work, we use the CAM assay to test the effect of the PPRHs against *ERBB2* either alone or in combination with trastuzumab in vivo.

The chicken CAM xenografts have been developed as an alternative for in vivo experiments with rodents, including their use as patient-derived xenografts (PDX) for precision medicine [27]. CAM assays display several advantages over mouse xenograft models, such as a much lower cost, a shorter timeframe, a suitable physiological environment, and good reproducibility. Additionally, they are aimed at experimenting in compliance with the Reduce, Replace, and Refine (3Rs) policy. The engraftment of tumor cell lines above the CAM is well-established, and it has been applied in pre-clinical screenings to assess the effect of anticancer drugs on tumor growth [28]. Furthermore, it has been shown that the CAM model is reliable for the analysis of metastatic spreading, angiogenesis, tumorigenesis, and drug sensitivity testing [29,30,31]. Transfection of breast cancer cells with the PPRHs against *ERBB2* prior to engraftment led to a significant decrease in tumor growth, which correlated with the reduction of HER-2 protein levels in those tumors, suggesting that the PPRHs silencing effect is enough to reduce the extra signaling of proliferation caused by the overexpression of HER-2 in breast cancer cells. The significant decrease in tumor growth obtained in samples transfected with HpI6 validates the usage of the PPRH technology as a gene silencing tool for breast cancer in vivo and reinforces the need *for* in vivo assays to confirm the results previously observed in vitro. In this direction, it is worth noting that the effectiveness of HpI6 is greater in an in vivo setting than in in vitro approaches. The same conclusion can be reached for the treatments with trastuzumab, which led to a decrease in tumor weight superior to what could be expected from the in vitro assays. Additionally, and in accordance with the in vitro assays, the effectiveness in vivo of HpI4 on reducing HER-2 protein levels was greater than that of HpI6, even though both PPRHs caused a significant reduction in tumor growth. Taking all of our results together, our results in the CAM assays indicated that they provide a more suitable physiological environment to evaluate the antitumor effects of both approaches. 

In conclusion, this work demonstrates that PPRHs are effective for inhibiting *ERBB2* gene expression both in vitro and in vivo, and this strategy can reduce cancer cell viability and tumor growth. This work represents a preclinical proof of principle for the in vivo application of PPRHs, opening the possibility to use this technology as a new therapeutic approach. Combinations of PPRHs and antibodies or other strategies, such as chemotherapy, could be a potential tool for the future treatment of breast cancer. Our data are encouraging, but it is important to note that pre-clinical CAM assays are far from suitable to extrapolate results to humans since the short experimentation time limits long-term observations, and further work is needed to improve the in vivo models and tests to be carried out before envisaging a future clinical performance of the PPRH technology.

## 4. Materials and Methods

### 4.1. Oligonucleotides

The Triplex-Forming Oligonucleotide Target Sequence Search tool (http://utw10685.utweb.utexas.edu/tfo/ MD Anderson Cancer Center, The University of Texas. Accessed 11 April 2023) [32] was used to locate the polypurine tracks present in the *ERBB2* gene and therefore the polypyrimidine targets to design the corresponding PPRHs. Table 4 describes the sequences and the names of the oligonucleotides used in this work.

For the design of the PPRHs against *ERBB2*, we chose among those sequences that were around 20–25 nucleotides in length and contained no more than three polypyrimidine interruptions to ensure a good formation of the hairpin by reverse Hoogsteen bonds. One of the arms of polypurines in the PPRH binds perfectly to the target DNA by Watson-Crick bonds without mismatches, allowing an interaction with high affinity [12]. Finally, we performed BLAST analyses to confirm the specificity of the selected PPRH sequences. 

PPRHs were synthesized as oligodeoxynucleotides without modifications and purified with gel filtration chromatography by Merck Life Science S.L.U. (Haverhill, UK). They were dissolved in a sterile Tris-EDTA buffer (1 mM EDTA and 10 mM Tris, pH 8.0) at 100 µM and stored until use at −20 °C. 

### 4.2. Cell Culture

The cell lines SKBR-3 and MDA-MB-453 of human breast cancer were obtained from the cell bank at the University of Barcelona (UB). Cell lines were validated by single-locus short tandem repeat typing (Bio-Synthesis, Inc., Barcelona, Spain) before their use and routinely tested for mycoplasma contamination. Both cell lines were used at low passage numbers (P < 20) for consistency between experiments.

Cells were maintained in a humidified 5% CO_2_ atmosphere in Ham’s F12 medium supplemented with 10% fetal bovine serum (both from GIBCO, Fisher Scientific S.L., Madrid, Spain) at 37 °C. The cells were detached for harvesting and expansion using 0.05% trypsin (Merck Life Science S.L.U., Madrid, Spain). 

### 4.3. Transfection of PPRHs 

Transfections were performed in 6-well dishes using 1,2-Dioleoyl-3-trimethylammonium propane (DOTAP, Biontex, Munich, Germany), by mixing the appropriate amounts of DOTAP and PPRHs, always maintaining a 1:100 molar ratio of PPRH:DOTAP, in serum-free medium up to 200 µL. The DOTAP/PPRH complexes were added after 20 min of incubation at room temperature to the cells that had already been plated in 800 µL of 10% FBS-containing medium to reach a final volume of 1 mL. When analyzing the combinations of PPRHs and trastuzumab, the antibody was added just upon cell transfection.

### 4.4. Fluorescent Microscopy and Flow Cytometry

Cells were plated in F12 medium in 6-well dishes and transfected the following day with 10–40 µM of DOTAP and 100–400 nM of a negative control PPRH labeled with fluorescein (FAM). Twenty-four hours after transfection, pictures of the cells were taken with a ZOE Fluorescent Cell Imager (Bio-Rad Laboratories, Inc., Spain). For flow cytometry analyses, cells were harvested by trypsinization, collected in PBS, and centrifuged at 800× *g* at 4 °C for 5 min. The pellet was resuspended in 400 μL of cold PBS. Propidium iodide was added to a final concentration of 5 µg/mL (Merck, Madrid, Spain) before flow cytometry analyses, which were performed in a Gallios flow cytometer (Beckman Coulter, Inc., Barcelona, Spain) by detecting the green and orange fluorescence of the control and treated cells. 

### 4.5. MTT Assay and Synergism Analyses

Cells were plated in 6-well dishes in F12 medium. Three to five days after transfection, 3-(4,5-dimethylthiazol-2-yl)-2,5-diphenyltetrazolium bromide (at 0.63 mM final concentration) and sodium succinate (100 μM) (both from Sigma-Aldrich, Madrid, Spain) were added to the culture medium and incubated for 3 h at 37 °C. Then, the culture medium was removed, and 1 mL of lysis solution (0.57% acetic acid, 10% SDS in DMSO) (Sigma-Aldrich, Madrid, Spain) was added. Absorbance was measured at 570 nm in a Varioskan^TM^ Lux multimode microplate reader (Thermoscientific, Madrid, Spain). Cell viability results were expressed as the percentage of cell survival compared to the controls transfected with a negative control.

The evaluation of the degree of synergism between the PPRHs and trastuzumab was performed using the CompuSyn software (v 1.0) [33]. For each combination, the Combination Index (CI) was calculated, and the final effect was classified depending on this value: CI < 0.3 Strong Synergism, CI 0.3–0.7 Synergism, CI 0.7–0.85 Moderate Synergism, CI 0.85–0.90 Slight Synergism, CI 0.90–1.10 Nearly Additive, and CI > 1.10 Antagonism. 

### 4.6. RNA Extraction and Analysis

Total RNA was extracted using Trizol reagent (Life Technologies, Madrid, Spain) following the instructions of the manufacturer. RNA concentration was quantified in a Nanodrop ND-1000 spectrophotometer (Thermo Scientific, Wilmington, DE, USA) by measuring its absorbance (260 nm). Complementary DNA was synthesized from 1 µg of total RNA, 0.5 mM of each deoxyribonucleotide triphosphate (dNTP; Epicentre, Madison, WI, USA), 250 ng of random hexamers (Roche, Barcelona, Spain), 10 mM dithiothreitol, 200 units of Moloney murine leukemia virus reverse transcriptase (RT), 20 units of an RNase inhibitor, and 4 µL buffer (5×) (all three from Lucigen, Middleton, WI, USA). The reaction was incubated in a 20 µL reaction mixture at 42 °C for 1 h. 

*ERBB2* mRNA TaqMan probe (Hs01001580_m1; Life Technologies, Barcelona, Spain) was used to determine HER-2 mRNA levels, and Cyclophilin (*PPIA*) mRNA TaqMan probe (Hs04194521_s1; ThermoFisher Scientific, Madrid, Spain) was used as the endogenous control. The reaction was carried out in 20 μL containing 1× TaqMan Universal PCR Mastermix (Applied Biosystems, Madrid, Spain), 0.5× TaqMan probe, and 3 μL of cDNA. The conditions for PCR were 10 min of denaturation at 95 °C followed by 40 cycles of 15 s at 95 °C and 1 min at 60 °C using a QuantStudio 3 Real-Time PCR System (Applied Biosystems, Barcelona, Spain). mRNA quantification was performed using the ΔΔCt method, where Ct is the threshold cycle that corresponds to the cycle when the amount of amplified mRNA reaches the fluorescence threshold. 

### 4.7. Western Blot Analyses 

Total cell extracts were prepared in a lysis buffer (50 mM HEPES ((4-(2-hydroxyethyl)-1-piperazineethanesulfonic acid), 0.5 M NaCl, 1.5 mM MgCl_2_, 1 mM EGTA (ethylene glycol-bis(β-aminoethyl ether)-N,N,N′,N′-tetraacetic acid), 10% glycerol, 1% Triton X-100, pH 7.2) added with Protease Inhibitor Mixture (Merck Life Science S.L.U., Madrid, Spain) from either control cells or transfected with the PPRHs. The samples were maintained on ice for 1 h at 4 °C with vortexing every 15 min. After removing cell debris by centrifugation (10,000× *g* for 10 min), the concentration of protein was determined in the supernatants using a BioRad protein assay (based on the Bradford method), using bovine serum albumin as a standard (Merck Life Science S.L.U., Madrid, Spain).

Protein extracts from pooled tumors were prepared by homogenizing the samples (50 mg of tumor tissue) in 300 µL of RIPA buffer (50 mM Tris-HCl, 150 mM NaCl, 1 mM EDTA, 1% Igepal, 0.5% sodium deoxycholate, 0.1% SDS, pH 8.0) using a Polytron homogenizer PT 1200E and a 3 mm diameter probe (both from Kinematica ^TM^, Malters, Switzerland) at 20,000 rpm for three periods of 20′′ on a water-ice bath. Immediately afterwards, samples were centrifuged at 10,000× *g* and supernatants were kept until use at −80 °C.

Total protein extracts (75–100 µg) were separated electrophoretically on SDS–7% polyacrylamide gels and afterwards transferred to polyvinylidene difluoride membranes (Immobilon P, Millipore, Madrid, Spain) using a semidry electroblotter. Blots were incubated with antibodies against HER-2 (1:100 dilution; Neu (C-18); sc-284; Santa Cruz Biotechnologies, Heidelberg, Germany) or tubulin (1:100 dilution; CP06, Calbiochem, Merck, Darmstadt, Germany) to normalize the results. The signals were detected with HRP-conjugated anti-rabbit antibodies (1:2500 dilution; P0399, Agilent Technologies, Singapore) for HER-2 or anti-mouse antibodies (1:2500 dilution; sc-516102, Santa Cruz Biotechnology, Heidelberg, Germany) for tubulin and enhanced chemiluminescence using ECL^TM^ Prime Western Blotting Detection Reagent (GE Healthcare, Barcelona, Spain). Chemiluminescence was detected with Image-Quant LAS 4000 Mini (GE Healthcare, Barcelona, Spain), and results were quantified by densitometry using the Image-Quant TL software v 10.2 (ThermoFisher Scientific, Barcelona, Spain).

### 4.8. CAM Assays

MDA-MB-453 cells (330,000) were plated in 6-well dishes in F12 medium. The next day, transfection was performed using 30 or 40 µM DOTAP and 300 or 400 nM of PPRHs, respectively. After transfection, cells were incubated for 48 h in the presence or absence of 100 µg/mL of trastuzumab (Kanjinti, Amgen, Thousand Oaks, CA, USA). Then, cells were collected by trypsinization in 1 mL of F12 medium and centrifuged at 1200× *g* at 4 °C for 5 min. Cell pellets were resuspended in 25 µL of ice-cold PBS^++^ (supplemented with 1 mM CaCl_2_ and 0.5 mM MgCl_2_) and kept on ice until their inoculation in chicken chorioallantoic membranes (CAM). For this purpose, premium specific pathogen-free (SPF), fertile, 9-day-incubated embryonated chicken eggs were used (supplied by Gibert farmers in Tarragona, Spain). MDA-MB-453 cells diluted in PBS^++^ and matrigel (1:1) were inoculated on CAMs, and tumors were grown for 7 days [15,34,35]. On day 7 after inoculation, the tumors were excised, weighed, and immediately frozen in N_2_. A summary of the main steps of the entire process is depicted in Figure 5a. 

### 4.9. Statistical Analyses

Statistical analyses were performed using GraphPad Prism 9 (GraphPad Software v 9.0, Boston, MA, USA). Data are the mean ± SEM from at least three independent experiments or samples. The levels of statistical significance were denoted as follows: *p* < 0.05 (*), *p* < 0.01 (**), *p* < 0.001 (***), or *p* < 0.0001 (****). 

## Figures and Tables

**Figure 1 ijms-24-07073-f001:**
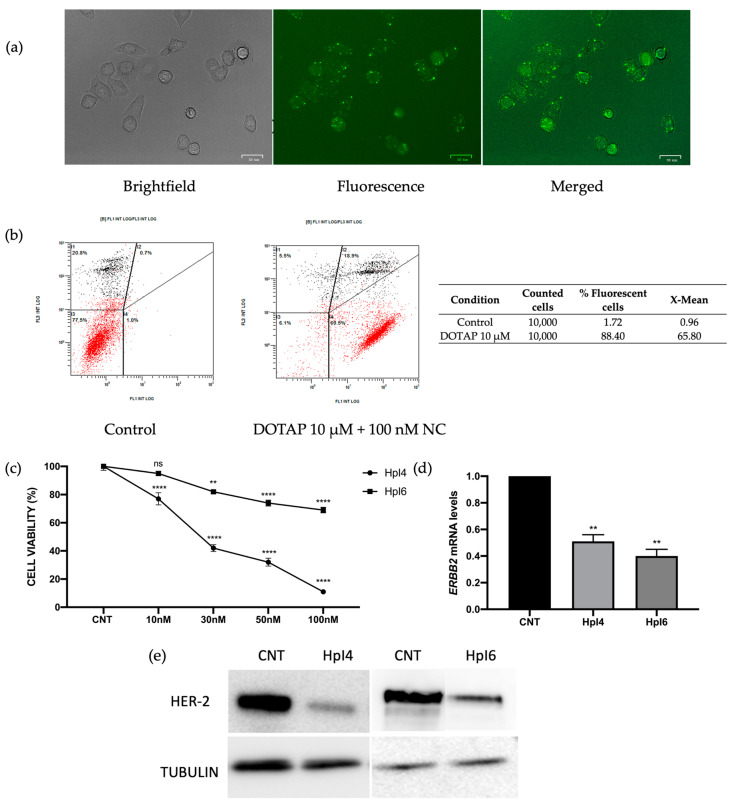
Uptake of the PPRHs in SKBR-3 cells was analyzed by fluorescent microscopy (**a**) and flow cytometry (**b**). 90,000 cells/well were plated and analyzed 24 h after transfection with a negative control of PPRH labeled with fluorescein (FAM). In (**b**), the intensity of green fluorescence (FL1) corresponding to transfected cells vs. the intensity of orange fluorescence corresponding to dead cells (FL3) is shown, and the results represent data from three independent determinations. (**c**) Dose-response to HpI4 and HpI6 in SKBR-3 cells (30,000 cells/well). The cell viability assay was performed 4 days after transfection. Results are expressed as a % of the control cells transfected with the negative control (NC). Results are expressed as a % of viability compared to cells transfected with DOTAP only, which corresponds to the control (CNT). (**d**) ERBB2 mRNA levels determined by RT-qPCR in SKBR-3 cells upon transfection of 100 nM of HpI4 or HpI6 for 24 h. In (**c**,**d**), results represent the mean +/− SEM from at least three experiments. Statistical significance was determined using a one-way ANOVA with Dunnett’s multiple comparison test (** *p* < 0.01, **** *p* < 0.0001, ns non-significant). (**e**) Representative image of HER-2 receptor protein levels analyzed by Western blot in SKBR-3 protein extracts from control cells or after 48 h of transfection with 100 nM of HpI4 or HpI6. The signal corresponding to tubulin was used to normalize the results. The determination was performed three times with consistent results.

**Figure 2 ijms-24-07073-f002:**
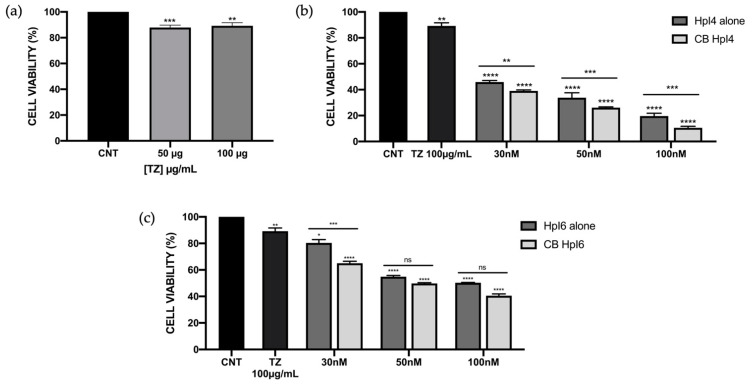
Effect on SKBR-3 cell viability of HpI4, HpI6, trastuzumab, and their combinations at different concentrations. (**a**) Effects of 50 and 100 µg/mL of TZ compared to control cells (non-treated). (**b**,**c**) Effect on cell viability of HpI4 (**b**) or HpI6 (**c**) at 30, 50, and 100 nM alone and in combination with 100 µg/mL of TZ. Cell viability assays were conducted with 60,000 plated cells 3 days after transfection. The data represents the mean +/− SEM from three or four experiments. Statistical significance was determined using a one-way ANOVA with Dunnett’s multiple comparison test (* *p* < 0.05, ** *p* < 0.01, *** *p* < 0.001, **** *p* < 0.0001, ns non-significant).

**Figure 3 ijms-24-07073-f003:**
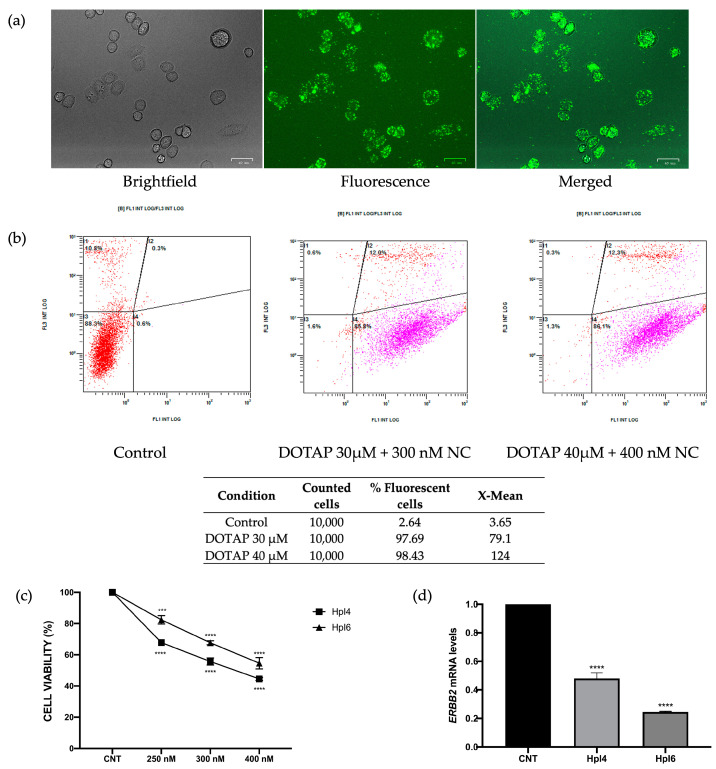
Cellular uptake of a fluorescent negative control PPRH (NC) in MDA-MB-453 cells (330,000 cells/well) was analyzed by fluorescent microscopy (**a**) and flow cytometry (**b**), 24 h after transfection with either 300 or 400 nM of the PPRH labeled with fluorescein (FAM) and 30 or 40 µM of DOTAP, respectively. In (**b**), the intensity of green fluorescence corresponding to transfected cells (FL1) vs the intensity of orange fluorescence corresponding to dead cells (FL3) is shown, and the results represent data from three independent determinations. (**c**) The dose-response effect of both intron-targeting PPRHs transfected at the indicated concentrations on MDA-MB-453 cell viability (330,000 cells/well) was determined 3 days after transfection. For each PPRH concentration, viability was calculated with respect to cells transfected only with DOTAP, which was considered the control. (**d**) Effect on ERBB2 mRNA levels upon PPRHs transfection in MDA-MB-453 cells (330,000 cells/well) for 24 h. Data from (**c**,**d**) represent the mean +/− SEM from three experiments. Statistical significance was determined using a one-way ANOVA with Dunnett’s multiple comparison test (*** *p* < 0.001, **** *p* < 0.0001).

**Figure 4 ijms-24-07073-f004:**
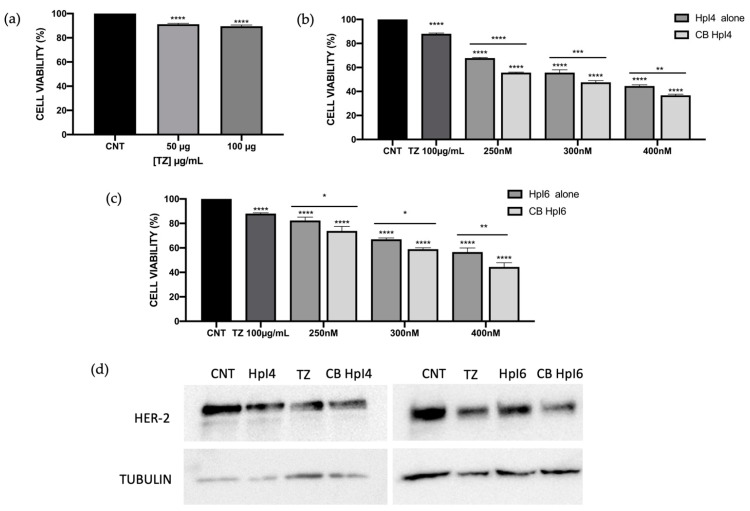
Effect on cell viability of HpI4, HpI6, trastuzumab, and their combinations at different concentrations in MDA-MB-453 cells. (**a**) Effect of 50 and 100 µg/mL of TZ compared to control cells (non-treated). (**b**,**c**) Effects of HpI4 (**b**) and HpI6 (**c**), either alone or in combination with 100 µg/mL of TZ, on MDA-MB-453 (330,000 cells/well) cell viability. All analyses were performed 3 days after transfection. Data represents the mean *+/−* SEM from at least three replicates. Statistical significance was determined using a one-way ANOVA with Dunnett’s multiple comparison test (* *p* < 0.05, ** *p*< 0.01, *** *p* < 0.001, **** *p* < 0.0001). (**d**) A representative image of the effect of HpI4 and HpI6 and their combinations with TZ on HER-2 protein levels. Protein extracts from MDA-MB-453 cells (330,000/well) were obtained after 48 h of transfection with 300 nM of PPRHs, either in the absence or in the presence of 100 µg/mL of TZ and analyzed by Western blot. The signal detected for tubulin was used to normalize the results. The determination was performed three times with consistent results.

**Figure 5 ijms-24-07073-f005:**
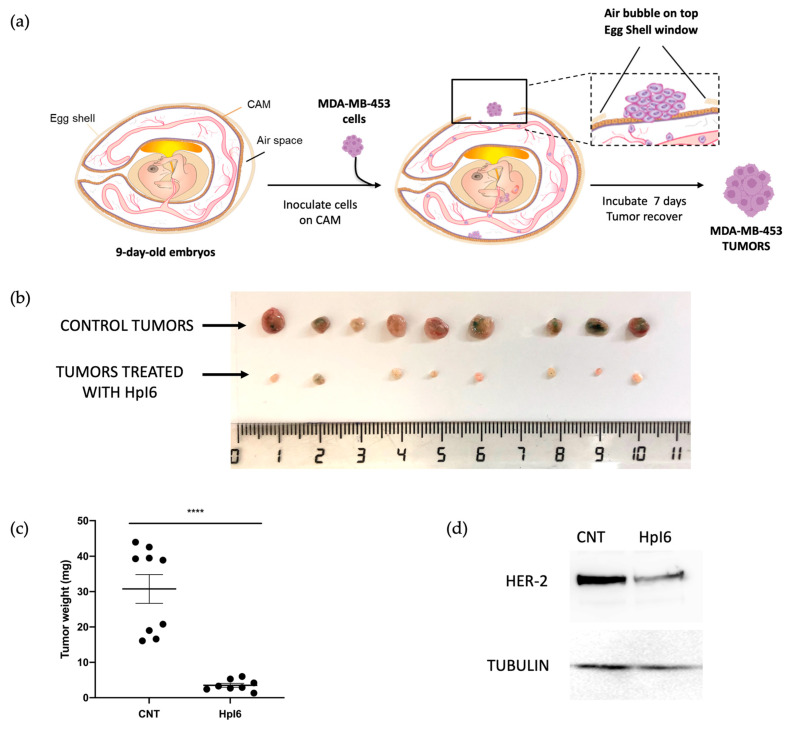
In vivo effects of HpI6. A sample of 330,000 MDA-MB-453 cells/well was seeded in 6-well dishes and transfected the following day with 400 nM of HpI6. Forty-eight hours after transfection, cells were inoculated in the CAM of chicken embryonated eggs. One week later, tumors were extracted and weighed. (**a**) Main steps in the CAM assay since the inoculation of MDA-MB-453 cells in the chick chorioallantoic membrane until the obtention of tumors. (**b**) The size of the control tumors (upper row) and those derived from cells transfected with HpI6 (lower row) is shown. (**c**) Distribution of tumor weight between control and transfected samples. (**d**) Representative blots of HER-2 levels determined by Western blot in total protein extracts prepared from pools of control (*n* = 9) and transfected tumors (*n* = 8). The signal corresponding to tubulin was used to normalize the results. Statistical significance was determined using an unpaired *t*-test (**** *p* < 0.0001).

**Figure 6 ijms-24-07073-f006:**
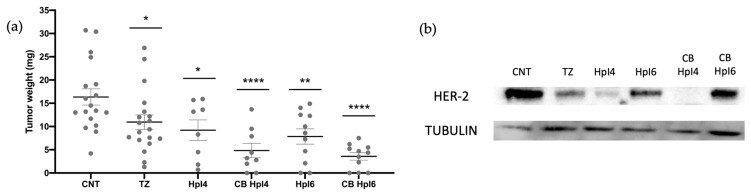
(**a**) Effect of HpI4, HpI6, trastuzumab, and their combinations on tumor weight. MDA-MB-453 cells were transfected with 300 nM of HpI4 or HpI6, either in the absence or presence of 100 µg/mL of TZ. Forty-eight hours after transfection, cells were inoculated in the CAM of chicken embryonated eggs, and one week later, tumors were extracted and weighed. The distribution of tumor weights between control, antibody-treated, and transfected samples is shown. (**b**) HER-2 levels determined by Western blot in total protein extracts prepared from tumor pools derived from control (*n* = 18), antibody-treated (*n* = 19), and PPRH-transfected cells, either in the absence (*n* = 8 for HpI4 and *n* = 11 for HpI6) or in the presence of TZ (*n* = 9 for CB HpI4 and *n* = 11 for CB HpI6). The tubulin signal was used to normalize the results. Statistical significance was determined using a one-way ANOVA with Dunnett’s multiple comparison test (* *p* < 0.05, ** *p*< 0.01, **** *p* < 0.0001).

**Table 1 ijms-24-07073-t001:** Analysis of synergy for the combinations of PPRHs and trastuzumab in SKBR-3 cells.

Dose HpI4 (nM)	Dose HpI6 (nM)	Dose TZ (µg/mL)	CI Media	Description
30	-	100	0.76	Moderate synergism
50	-	100	0.72	Moderate synergism
100	-	100	0.51	Synergism
-	30	100	0.66	Synergism
-	50	100	0.60	Synergism
-	100	100	0.93	Nearly additive

The concentrations of PPRHs and antibodies, the calculated combination index (CI), and the description of the resulting effect are shown.

**Table 2 ijms-24-07073-t002:** Analysis of synergy for the combinations of PPRHs and trastuzumab in MDA-MB-453 cells.

Dose HpI4 (nM)	Dose HpI6 (nM)	Dose TZ (µg/mL)	CI Media	Description
250	-	100	0.80	Moderate synergism
300	-	100	0.81	Moderate synergism
400	-	100	0.86	Slight synergism
-	250	100	0.90	Slight synergism
-	300	100	0.81	Moderate synergism
-	400	100	0.85	Moderate synergism

The concentrations of PPRHs and antibodies, the calculated combination index (CI), and the description of the resulting effect are shown.

**Table 3 ijms-24-07073-t003:** Analysis of tumor weights from CAM assays.

Condition	Mean Weight (mg)	SEM	N	% Reduction vs. Control
Control	16.35	1.75	18	-
TZ 100 µg/mL	10.96	1.56	19	32.97
HpI4 300 nM	9,21	2.21	8	43.65
CB HpI4	4.83	1.54	9	70.44
HpI6 300 nM	7.86	1.65	11	51.96
CB HpI6	3.57	0.80	11	78.15

The different experimental groups, mean tumor weight for each group, SEM, number of samples in each group, and % of tumor weight reduction compared to the control group are shown.

**Table 4 ijms-24-07073-t004:** PPRH sequences used in this work.

PPRH Name	Sequence from 5′ to 3′	Target Location in *ERBB2*
HpHer2-I4-T	GGGAGAGGGAGTGGGAACAGAGTGGG (T)_5_ GGGAGAGGGAGTGGGAACAGAGTGGG	Intron 4
HpHer2-I6-C	AGATGAGAGGTGAGAAGGAAGGAGAGAG (T)_5_ AGATGAGAGGTGAGAAGGAAGGAGAGAG	Intron 6
Negative control	AAGAAGAGAAGAAGAAGAA (T)_5_ AAGAAGAAGAAGAGAAGAA	None

The name, sequence, and position of the PPRHs specifically designed against the *ERBB2* gene and the sequence used as the negative control are shown. C and T refer to the types of PPRHs designed, either against the template (T) or coding (C) strands.

## Data Availability

All data are presented within the submitted manuscript.

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
