# Peer review of "In Vitro and In Vivo Effects of the Combination of Polypurine Reverse Hoogsteen Hairpins against HER-2 and Trastuzumab in Breast Cancer Cells"

_ijms, 2023, doi:10.3390/ijms24087073_

Round 1
Reviewer 1 Report (Previous Reviewer 1)
In this revised manuscript, the authors performed new experiments which supports their conclusion and greatly improved the quality of the manuscript. The authors addressed the concerns appropriately therefore, in my opinion this manuscript is suitable for publication.
Author Response
Thank you very much for your comments.
Reviewer 2 Report (New Reviewer)
The authors describe the development of two oligonucleotides (PPHRs) targeted for silencing an oncogene, ERBB2, that is overexpressed in various types of cancers. The authors utilized two types of breast cancer cell-lines to test the effect of each PPHR individually and in combination with Trastuzumab, an anti-body-based cancer drug. Their results demonstrate that the PPHRs of interest, both are capable of suppressing ERBB2 and a synergistic effect was reported when certain doses of the PPHRs were co-administrated with Trastuzumab. Furthermore, the authors used CAM as an in vivo model to confirm their results and have reported a reduction in the tumor size in response to the treatments.
General Comments
Study design and methodology:
1) What is the rationale for choosing to study the effect of the PPHRs of interest in combination with antibody-based therapy (Trastuzumab) as opposed to more conventional cytotoxicity-based/chemotherapy?
2) At what passage number the cell lines were used?
3) Where the cells tested for mycoplasma prior to conducting the experiments?
4) The indicate cell seeding density refers to number of cells/cm2 or cells/well of a standard 6-wells plate?
5) Is there a reason for using the negative control PPHR as a proxy to estimate the oligonucleotide uptake in cells rather than doing the assessment directly on HpHer2-I4-T/HpHer2-I6-C ?
6) Which fluorophore was used to label the PPHRs?
7) During the post transfection period, are the cells returned to serum-free or serum-containing growth media.
8) Adding a figure illustrating the CAM assay steps would be very beneficial to the reader
9) The number of biological replicates used in each experiment have to be clearly indicated in the figure legend for each panel.
Results:
1) Figures 1/2(a):
- The morphology of the cells in the phase-contrast images indicates spares and sub-optimal health of the cells.
- The fluorescence signal shown in the representative image is not satisfactory. The image shows a faint background and there is not genuine signal? It would be concerning if all experiments in SKBR-3 display the same quality of signal?
- The number of biological replicates (n) used in this experiment is not indicated
2) Figures 1/2(b): what does the X and Y axis represent? and which fluorophore(s) are being detected by the FC
3) Figure 1(c): the stats are missing for this figure
4) Figure 1(e): The number of biological replicates (n) used in this experiment is not indicated
5) Lines 264-273: It would be easier for the reader to follow if the effect of the treatment(s) protocols (percentages) were expressed in terms of cell viability rather than cytotoxic effect to match the corresponding figures and the same applies in lines (305-312)
6) Table 2 and Lines (268-269) “As shown in Figure 2b and 2c, in all cases, the combination of each PPRH with the antibody was more effective than either the PPRH or the antibody alone” The conclusion described in this statement requires revision. The combined treatment of Hpl6 yielded statistically significant results -compared to Hpl6 alone-only at the concentration of 30nM, therefore synergistic effect can’t be concluded for the higher doses.
7) Figure (2c):
- The stats are missing for this figure
- Why did the authors normalize cell viability for MDA-MB-453 to cell transfected with DOTAP-only but for SKBR-3 (Figure 1c) cells were normalized to untransfected (control) cells?
8) Why protein expression of HER-2 in response to PPRHs single treatment was not assessed in MDA-MB-453 as in Figure 1e
9) Why protein expression of HER-2 in response to PPRHs combined treatment was only assessed in MDA-MB-453 but not in SKBR-3.
10) The authors are encouraged to omit the part on SKBR-3 cells, since more data was collected from MDA-MB-453 and the latter cell line was the one xenografted into the CAM system.
11) The authors are encouraged to apply densometric analysis on the western blots (particularly for the combined treatments experiments) to be able to detect the effects across and between conditions.
12) This statement “The signal corresponding to Tubulin was used to normalize the results.” Could be changed to “Tubulin was used as a loading control” unless densometric analysis is going to be included.
13) Drugs abbreviations in the figures need to be unified for example: Hpl6 is sometimes written as I6, Trastuzumab, sometimes written as Ab.
14) In the mRNA expression experiments figures HER2 should be replaced with the official gene symbol ERBB2.
15) Figure 5c: - can the authors include images of CAM tumor treated with Hpl4 in comparison to control
-Why HER-2 was not evaluated in response to Hpl4 as well? and what is the number of biological replicates used here?
16) Figure 6:
-what is the reason for not conducting western blot analysis on these samples?
- stats-test results are missing
17) The authors did not discuss the study limitations and how they propose to advance this line of research
Round 2
Reviewer 2 Report (New Reviewer)
Point 9. The number of biological replicates from each western blot experiment was derived, is still missing from the corresponding figure legends
Point 16. Fig 1c/3c cell viability plots, the provided explanation is unclear, did the authors mean to say that for both cell lines viability was compared to DOTAP+NC and not untransfected cells? if yes could this be stated plainly in the text to avoid confusing the reader
Point 18. The authors response to this point could be inserted where relevant in the text
Point 20. The results of the densometric analysis and the accompanying stats are not included in the revised version
Point 24. The provided explanation is not convincing. The authors already show data from CAM-tumors treated with Hpl4, we asked here for the authors to display images of the tumors excised and to present western blot experiments results as they have done for Hpl6 in Figure 5.
Point 25. Figure 6 c is missing along with the statistical analysis referred to in the authors response
Round 3
Reviewer 2 Report (New Reviewer)
most of the comments were addressed
This manuscript is a resubmission of an earlier submission. The following is a list of the peer review reports and author responses from that submission.
Round 1
Reviewer 1 Report
In the current context of the study topic, the article entitled “In vitro and in vivo effects of the combination of Polypurine Reverse Hoogsteen Hairpins against HER-2 and Trastuzumab in breast cancer cells” is interesting. This manuscript describes the effect of two Polypurine Reverse Hoogsteen (PPRH) hairpins directed against the ERBB2 gene which is overexpressed in positive HER-2 breast tumors. The combination of these specific PPRHs with Trastuzumab was also explored in breast cancer cell lines both in vitro and in vivo. Authors observed that PPRHs designed against two intronic sequences of the ERBB2 gene decreased the viability of SKBR-3 and MDA-MB-453 breast cancer cells. The decrease in cell viability was associated with a reduction in HER-2 mRNA and protein levels. They observed also that in combination with Trastuzumab, PPRHs showed a synergic effect in vitro and reduced tumor growth in vivo. The results obtained in this study have a promise for future investigation of PPRHs as a therapeutic tool for breast cancer. As such, it is suitable for publication in International Journal of Molecular Science in its present form, however, prior to acceptance, the manuscript needs some changes in order to improve the quality of the manuscript.
1-The authors should include in the Materials and Methods section if they authenticate the cell lines confirming that cell lines are derived from the correct species and donor, and that they are contamination-free (mycoplasma test).
2-In figures 1a and 3a the uptake of the PPRH-liposome-complex is not clearly observed due to the large amount of background fluorescence. The authors should improve these images by reducing the background fluorescence.
3-In figure 1e the intensity of the charge control is not the same in the control and in the sample treated with HpI4, so it cannot be concluded that there is a decrease in the Her2 protein. Authors should repeat the western blot loading the same amount of protein in both lanes. In addition, the quality of the signal from the tubulin western blot shown in the figure is quite poor. The authors should improve it.
Reviewer 2 Report
In this manuscript authors describe design, synthesis and biological investigation of the series of oligodeoxynucleotides (ODNs), called polypurine reverse Hoogsten hairpins (PPRHs). The manuscript is somewhat confusing, besides the fact that it was submitted in a still edited version, so it is not clear if this is the final version. Introduction is a strong self-promotion with first 12 references to self-citation, so it is clear that this group almost exclusively use this methodology and although there are number of papers published before, all are from one group, surprising. Anyway, the concept is not a bad idea, but the way the experiments were designed and executed and data is presented, this is below the borderline of acceptance. There are some major point and minor corrections below, should the editor decide to accept it for publication.
Major and minor comments:
The length of scramble sequence fragment is different from the target fragment. Besides, there is no T in the “scramble” (HpSc9) negative control, as it is in HpHer2, so it is not a scramble sequence at all.
P3L111 “Transfections were performed [...], by mixing the appropriate amounts of DOTAP and PPRHs, always maintaining a 1:10 ratio PPRH: DOTAP,” How the ratio was estimated? is it a molar ratio, weight ratio or other? If this is a molar ratio then 1:10 does not allow the full charge screen and more importantly the charge ratio between HpSc9 (41nt) and HpHer2 samples (61nt) and DOTAP is different.
P6L222 “100 nM of PPRH transfected with 10 μM DOTAP”: this is a ratio 1:100 not 1:10 as in Materials and Methods.
Figure 2b Results from flow cytometry seem to be also confusing: the number of counts (cell’s events, Figure 2b and Figure 3b) is not provided so it is not clear whether it is equal for control and sample experiment.
Figure 2e the difference in intensity level for tubulin disqualifies the result for the HER-2 level in the Hpl4 experiment.
P7L258-59 “[...] SKBR-3 cells were transfected with either HpI4 or HpI6 in the presence of 100 μg/mL of Trastuzumab,” What was the transfection reagent? DOTAP, Lipofectamine or other. If these are DOTAP liposomes, the 1:10 molar ratio (page 3) might be not sufficient to screen all ODN charges and counterpart (complex) large antibody such as Trastuzumab and that is probably why the effect of antibody is negligible in most experiments here.
However, microscopy images are the most disturbing here. Cells in all images, including brightfield control (Figures 2a, 3a) look occluded, rounded and very unhealthy cells, there are just few of SK-BR-3 in the field. Actually, close inspection of images in Figure 2a disproves the claim, cause in the group of only healthy cells (right-down corner) the fluorescent particles are everywhere, except where cells are. The MDA-MB-453 cells (figure 3a) also look very miserable, few mostly dead cells in the view.
The only reasonably valid experiment seems to be the chicken chorioallantoic membrane assay, however, this is out of the reviewer’s filed of expertise and cannot be properly validated.
Minor comments:
P2L45 “Coding-PPRHs” selected fragments called PPRHs are not coding sequences.
P2L51 “PPRHs can decrease gene expression through different mechanisms,...” The mechanism under which the fragment called PPRHs influence the cell survival is not fully understood and that PPRHs regulate the gene expression is overstatement.
P2L58 “Sars-CoV2” should be SARS-CoV-2,
P3L92-94 The purity of oligodeoxynucleotides (ODNs) was not confirmed, nor revealed.
P6L221 “flow citometry" should be flow cytometry,